# Inoculation reduces social media engagement with affectively polarized content in the UK and US
Fintan Smith [1,2,6], Almog Simchon [1,3,6], Dawn Holford [1] & Stephan Lewandowsky [1,4,5]

The generation and distribution of hyper-partisan content on social media has gained millions of exposure across platforms, often allowing malevolent actors to influence and disrupt democracies. The spread of this content is facilitated by real users' engaging with it on platforms. The current study tests the efficacy of an 'inoculation' intervention via six online survey-based experiments in the UK and US. Experiments 1–3 (total $N = 3276$) found that the inoculation significantly reduced self-reported engagement with polarising stimuli. However, Experiments 4–6 (total $N = 1878$) found no effects on participants' self-produced written text discussing the topic. The implications of these findings are discussed in the context of the literature on polarisation and previous interventions to reduce engagement with disinformation.

In the last few decades, political polarisation has grown in numerous countries worldwide[1–3]. Of particular concern is the rise in affective polarisation, that is the disparity between feelings of warmth towards political in-groups versus political out-groups[4]. This concept of affective polarisation is rooted in Social Identity Theory[5], which posits that humans are naturally inclined to categorise themselves and others into in-groups and out-groups, with greater salience of these identities encouraging greater positive affect towards the in-group and greater animosity towards the out-group. However, rather than observing any great increase in in-group warmth, instead researchers have noted declining warmth towards political foes, or a growth in so called negative partisanship[6].

The rise in affective polarisation may in part be driven by our increased dependence upon social media for news gathering and the explosion in the number of hyper-partisan outlets (e.g. Breitbart) that create and spread hyper-partisan content on social media[7–10]. Much of this content could be classed as disinformation, the deliberate creation and distribution of false or manipulated information[7,8,11]. Creators of such content have varied objectives, including monetisation of the sharing of sensationalist or partisan news, but often there is an incentive to influence and reduce trust in democratic processes by increasing group polarisation[12–15]. Indeed, exposure to partisan reporting that is critical of those out-groups has been found to decrease ratings of trust and liking of those groups, feeding into negative partisanship[6,16].

The full scale of the problem has become clearer over the last decade. For example, investigations into the reach of foreign disinformation operations, including the internet research agency based in St Petersburg, have found hundreds of millions of exposures to disinforming and hyper-partisan posts on Twitter and Facebook during the 2016 EU membership referendum in the UK, and the 2016 US presidential election[14,15].

More recently, the impact of the spread of hyper-partisan disinformation was seen in the unfounded accusations of irregularities or fraud in the 2020 US election by the losing candidate, Donald Trump, which were amplified by the mass media and pushed on social media by senior supporters and members of the Trump administration. This campaign successfully convinced over 65% of Republican voters that the election was "stolen" from Donald Trump[17] and it later inspired a violent insurrection at the US Capitol building on January 6th, 2021, with which the deaths of seven people have since been connected[18,19].

The successful dissemination of hyper-partisan content via social media depends on real users' engagement with it[20]. Content curation algorithms prioritise content to appear in users' feeds that have greater numbers of reactions, shares, comments and clicks from other users, meaning that hyper-partisan social media content that receives more engagements from users can be prioritised above high-quality reporting on the same subject that receives relatively fewer engagements[21–23]. Social media users also tend to like and share partisan posts by politicians that are negative towards out-groups over posts positive towards in-groups[24]. Engagement with partisan content can serve as a form of endorsement cue, rewarding those who post and share it and therefore reinforcing the behaviour, whilst also lending hyper-partisan content, including disinformation, a sense of legitimacy to those exposed to it[25,26].

Understanding what underlies user engagement with such content has been the focus of a large amount of research over the last decade. One

[1]School of Psychological Science, University of Bristol, Bristol, UK. [2]YouGov PLC, London, UK. [3]Department of Psychology, Ben-Gurion University of the Negev, Beer Sheva, Israel. [4]Department of Psychology, University of Potsdam, Potsdam, Germany. [5]School of Psychological Science, University of Western Australia, Crawley, WA, Australia. [6]These authors contributed equally: Fintan Smith, Almog Simchon. ✉e-mail: stephan.lewandowsky@bristol.ac.uk

position holds that users may simply fail to pay attention to signs that flag content as inaccurate. Social media environments are information rich, which can overwhelm users' attentional capacity, leading to uncritical engagement with disinformation. In support, both lab studies and analyses of behavioural social media data find that those who score lower on the cognitive reflection test, a measure of reflective thinking, are more likely to both believe and share inaccurate content on social media[27–29]. In addition, accuracy nudge interventions, which prompt participants to assess the accuracy of content when they encounter it, significantly reduce sharing of inaccurate content, as well as respondent's belief in such content[30,31].

However, this explanation fails to elucidate why some people will knowingly share attitude congruent, often hyper-partisan, political disinformation. For example, ref. 30 found that in 16% of cases where inaccurate headlines were shared on social media, the sharer knew that the headline was inaccurate. Partisanship may explain this phenomenon. Re-analysing data from previous studies, ref. 32 found that the effectiveness of accuracy nudge interventions is significantly moderated by strength of partisanship, with no significant effect of otherwise effective accuracy nudge interventions amongst strongly identifying US Republicans, a group that were particularly likely to have shared disinformation in the run up to the 2016 US election[27].

Affective polarisation itself may motivate user engagement with hyper-partisan content. Previous work has demonstrated that partisans may be more motivated to *believe* and *share* fake news that is congruent with their political identity[33–35]. Indeed, political polarisation has been demonstrated to motivate citizens to engage in political expression in the form of protest[36]. Therefore, partisans may be likewise motivated to engage in easier and non-physically confrontational expressions of their political views, including expressions of outgroup hate, by sharing or posting hyper-partisan content online[37]. For instance, social media users are likely to respond with more toxic comments to political content from out-groups compared to in-groups[38]. Recent work suggests that partisan users experience a trade off between these partisan motivations—the political utility of using the content to promote in group love or out-group hate—and their motivations for accuracy, with the former often trumping the latter[39]. Testing this hypothesis directly with Twitter user data from US survey respondents, ref. 39 found that both stronger identification with a political party, as well as the extent of partisans negative partisanship, were associated with the sharing of fake news, while cognitive reflection score was not. Moreover, negative partisanship had twice the predictive power for sharing as in-group identification and was the most powerful predictor of political fake news sharing.

This desire among partisans to attack political out-groups, born out of negative partisanship, can also be seen in the political content that tends to receive engagement across social media platforms. Decisions to engage with disinformation on social media can be based on limited information, even just a headline[40]. Certain features of posts that predict engagement have been identified. For example, the inclusion of moral-emotional language appealing to one's sense of right and wrong increases the likelihood of sharing at a rate of 20% per word[41]. Other works suggest that these effects are further exaggerated if content is negative, particularly if expressing anger[42,43], though this could be context specific[41]. Emerging evidence suggests that out-group derogating language is also a driver. Partisans are more likely to share and believe in news that derogates opposing politicians[44]. Using data scraped from both Twitter and Facebook, ref. 45 found that use of out-group cues (e.g. "Clinton") strongly predicted angry reactions on Facebook, and the use of each of these cues in headlines increased the probability of posts being shared by 67%—an effect several times larger than that of moral-emotional language. Ref. 39 focused on all types of political news sharing in their analysis, finding that out-group derogating affectively polarised content in news headlines predicted sharing of *any* news content, not just fake news.

If partisan motivations to engage with hyper-partisan content often come out on top over our motivation towards accuracy, then a different approach to accuracy nudges is required to address these motivations. No online interventions have yet been developed that explicitly tackle partisan motivations to engage with hyper-partisan content. However, one of the most promising approaches that has been applied for tackling the

proliferation of disinformation and fake news in general is psychological inoculation, which seeks to build individuals' resistance against mis-informing messages that they are likely to encounter[46]. A typical inoculation consists of a warning about an imminent threat from misinforming messages, followed by examples of the threat[46,47]. Inoculation can be applied to a broad spectrum of messages, for example, technique-based inoculations seek to warn and equip people with knowledge of logical fallacies and manipulation techniques that are common features of disinforming content. Multiple lab and field studies confirm that inoculation, in the form of text, videos or games, can effectively increase people's ability to discern between fake and real news and reduce their engagement with misinformation on social media[48–50].

Unlike corrective interventions, such as labelling of disinforming posts online, inoculation does not require knowledge of specific content and rebuttals of that content which take time to prepare[51]. Instead, inoculation can be reasonably easily rolled out online as public-service adverts and could even be triggered in users' feeds when encountering content that is characteristically misinforming or highly partisan. For example, ref. 49 demonstrated that an inoculation video could be implemented effectively within the YouTube platform to achieve a highly scalable effect. In addition, this approach has the advantage of generalising across malicious content on different topics and from different perspectives. For example, a single technique-based video inoculation was able to decrease sharing of both far right Islamophobic and radical Islamist content online—an impressive demonstration of the broad reach of inoculation[52].

The current study tested a technique-based video inoculation intervention that aims to undermine partisan motivations to engage with hyper-partisan content typical of what users might encounter online, by highlighting the negative consequences of the spread of such content for democracy and society. Though much hyper-partisan content contains disinformation, the content we deal with in the current study has no explicit truth value, because factual status is irrelevant to the phenomenon under consideration, which is primarily affective polarisation[7,11]. The video inoculates against affectively polarised language, focusing on three manipulation techniques prevalent in misinforming posts, often applied to derogate out-groups (scapegoating, ad-hominem attacks and emotional language)[41,53].

We tested the intervention across six studies. Experiments 1–3 tested the intervention's efficacy at reducing self-reported engagement with affectively-polarised stimuli, i.e. hyper-partisan content that derogated the participants' political out-group. Experiments 4–6 tested the intervention's efficacy at reducing the use of affectively-polarised language in participants' written text produced post-intervention. The intervention was administered as a video in all except Experiment 3, where it was administered in text format. The video was originally developed for Experiment 1 and then enhanced for Experiment 2 onwards by integrating language from a polarisation dictionary[12] into the intervention. By doing so, we pursued a data-driven approach to inoculation in place of the theory-driven approach taken in Experiment 1. A large body of research supports the efficacy of theory-driven inoculation techniques against rhetorical devices of argumentation (e.g. incoherence, ad-hominem, scapegoating, etc). Here we propose a 'data-driven' technique: inoculating against the use of linguistic features that have been shown to reflect affective polarization[12]. Experiments 1 and 2 were centred around the 2016 referendum on the UK's membership of the EU, which remains an affectively polarised issue[54]. Experiments 3–6 were centred around the polarised abortion debate in the US[55,56].

Our primary hypotheses were that affectively polarised stimuli would receive higher engagement likelihood ratings overall; that our inoculation would significantly decrease engagement likelihood ratings for affectively polarised stimuli, and that there would be a significant interaction between stimuli type and video condition, such that any effect of the inoculation on engagement would apply only to the affectively polarised stimuli.

For brevity, we report only results pertaining to the primary hypotheses. Supplementary analyses are provided in the Supplementary Information. All experiments were pre-registered except Experiment 2, due to an

**Fig. 1 |** Example stimuli, showing posts employing ad-hominem attacks, displayed to Leavers (left) and Remainers (right) in experiment 1.

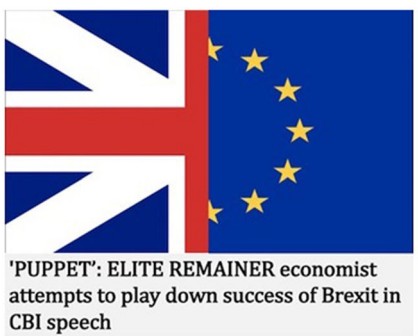

'PUPPET': ELITE REMAINER economist attempts to play down success of Brexit in CBI speech

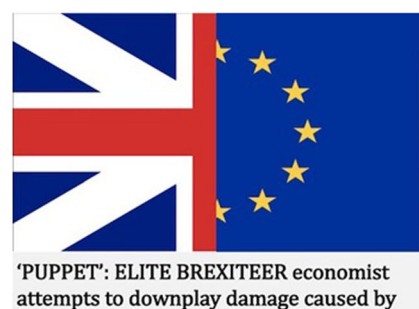

'PUPPET': ELITE BREXITEER economist attempts to downplay damage caused by BREXIT in CBI speech

administrative oversight, however Experiment 2 was intended as a replication of Experiment 1.

## Methods

Prior to all data collection, ethical approval was granted by the University of Bristol's Psychology Ethics Committee, approval nos.: 10176 and 11063. All studies were performed in accordance with the relevant guidelines and regulations approved by the Research Ethics Committee. Informed consent was obtained from all participants prior to their participation in the study. Studies 1, 3, 4, 5 and 6 were pre-registered with the Open Science Framework: https://osf.io/zakvj (pre-registered on February 26, 2022) and AsPredicted: https://aspredicted.org/rxvs-qnjg.pdf; https://aspredicted.org/blind.php?x=61Q_PPX (pre-registered on October 12, 2022), https://aspredicted.org/2x49-jmp8.pdf (pre-registered on August 5, 2024), https://aspredicted.org/q7yw-zg4p.pdf (pre-registered on August 14, 2024) and https://aspredicted.org/2scr-dcvv.pdf (pre-registered on September 9, 2024) All stimuli used in the experiments are available in Supplementary Tables S1–S3. We checked prior to analyses that the data met the assumptions of normality and variance for the statistical tests.

### Experiment 1

**Participants**. We planned for a target sample of 508, to achieve at least 462 participants after accounting for 'don't know' responses and response attrition. This target would have given >95% power at $\alpha = 0.05$ to detect a small within-between interaction ($f = 0.1$; [57]) in our planned ANOVA design Our sample size target was based on an a priori power analysis to detect small effects in a mediation analysis (reported in Supplementary Information). This indicated that 462 participants were required for 80% power.

Participants were recruited via a third-party online panel provider (YouGov) in March 2022. The study was included on three waves of a regular omnibus survey that employs quota sampling to recruit a politically representative sample of 1600 adults in Great Britain. Only participants who identified as a "Leaver" or a "Remainer" and reported having previously shared political content on social media (see measures) were included. Due to these stringent inclusion criteria, only around 300 participants per wave of data collection qualified. Our target sample size was achieved after the first two waves of data collection, however, due to an imbalance between Leavers and Remainers in the sample, we required a third wave, that only targeted Leavers. The final analysis included 760 participants (309 Leavers and 451 Remainers), with ages ranging from 18 to 87 ($M = 48.77$, $SD = 15.89$). Quality of responses was ensured by YouGov, which automatically excluded speeders (a completion time two or more standard deviations below the mean) and suspected bots (using geo-tagging and straight-line checks). See Supplementary Table S4 for sample characteristics.

**Design**. The experiment used a 2 (video condition: inoculation or control; between-subjects) ×2 (headline type: affectively polarised (derogatory) or non-derogatory; within-subjects) mixed design.

**Materials**. *Videos.* A single inoculation video was produced which sought to warn against partisan motivations for engaging with affectively polarised out-group derogating content by highlighting the negative consequences of spreading such content for democracy and society. The video focused on three manipulation techniques prevalent in misinforming posts, often applied to derogate out-groups (scapegoating, ad-hominem attacks and emotional language)[41,53]. In keeping with inoculation methodology, the video warns viewers of an imminent threat of manipulation and provides an example of posts that employ particular techniques[46,47]. The example texts used "people in red ties" and "people in blue ties" to represent two opposing political groups. The video can be viewed at https://inoculation.science/inoculating-against-affectively-polarised-content/.

For the control condition participants watched an existing animated informational video about the British political system (with permission from the creator)[58].

*Stimuli headlines.* We constructed 12 synthetic headlines around three themes on the topic of Brexit, namely 'failure to successfully execute Brexit', 'media bias' and 'economic consequences' (see Fig. 1). For each theme, two affectively polarised (derogatory) headlines, and two issue-polarised (non-derogatory) headlines were created. One of each type of headlines was congruent with the views of Remainers, and the other was congruent with the views of Leavers (see Supplementary Table S1). Each of the derogatory headlines used one of the three techniques highlighted in the inoculation video but did not contain factually incorrect information.

To ensure the synthetic stimuli were perceived as being derogatory, and therefore affectively polarised, to an out-group, we ran a validation study with a politically representative sample of 1565 adults in Great Britain via YouGov. Derogating, affectively polarised, stimuli were rated to be significantly more derogating than non-derogating stimuli (see Supplementary Information, Supplementary Table S5 and Supplementary Fig. S1).

*Measures.* Participants responded to the following screening questions, which were used as inclusion criteria for the experiment.

*Brexit identity.* Participants were asked whether they identified as a Leaver or a Remainer.

*Social media sharing.* We used a Likert-scale measure of how often participants share content, and subsequently political content, on social media.

We obtained the following socio-demographic measures.

*Age.* This was obtained from the panel provider and derived from date of birth provided by panellists.

*Brexit identity strength.* Based on a measure employed by previous studies on polarisation[59], this measure consisted of five sub-components measured on five-point Likert agree-disagree scales. Higher average scores indicate stronger 'Remainer' or 'Leaver' identity.

Participants completed four outcome measures for the experiment.

*Clicking likelihood.* Participants rated how likely they would be to click on the stimuli post presented to find out more about it. Ratings were provided on a four-point single Likert item with an additional 'not sure' option.

*Sharing likelihood*. Four-point single Likert item with a 'not sure' option that asks Participants rated how likely they would be to share the stimuli on social media.

*Reaction*. Participants selected an emotional reaction in an emoji format similar to that used by Facebook. This was an exploratory measure and the results are reported in the Supplementary Information and Supplementary Table S6.

*Affective polarisation* We used a feeling thermometer to measure levels of affective polarisation[60]. Participants rated their feeling towards the out-group on a temperature scale of 0–100 with increments of 1. Colder temperatures signal greater animosity towards the out-group. Given constraints of survey length, and our particular interest in negative partisanship, we did not include a measure of feelings towards participants' in-group.

**Procedure**. Figure 3 summarises the procedure. Eligible participants completed the socio-demographic questions and were then randomly allocated to see either the inoculation video or the control video. Participants then completed the polarisation measure. They then viewed a pair of headlines, each presented in sequence on a different page (with random order of presentation). One headline was derogatory and one was not. The headlines were always congruent with the participant's Brexit identity. The derogatory headline of the pair either used scapegoating, ad-hominem attacks, or emotional language as a technique. Participants were randomly allocated to see one of the techniques in their headline pair. For each headline, participants completed the reaction, clicking likelihood and sharing likelihood measures. See Supplementary Tables S7–S9 for all headline pairs.

## Experiment 2
**Participants**. Participants were again recruited via YouGov, with the same inclusion criteria as in Experiment 1. To replicate the same primary analyses of Experiment 1, we again pursued a sample size of 508 Leaver and Remainer participants who reported sharing political content on social media. In total, we ran four waves of data collection that recruited a total of 864 respondents (378 Leavers and 486 Remainers), aged 18–86 ($M = 50.64$, $SD = 15.72$) for our final analysis. Data collection took place over four periods of 24 h between early November 2022, and mid-December 2022. Quality of responses was ensured by YouGov as in Experiment 1. For sample characteristics, see Supplementary Table S10.

**Design**. The design was identical to that of Experiment 1.

**Materials**. *Videos*. We used the same control video as Experiment 1, but made a small alteration to the inoculation video to include words from the polarisation dictionary[12], which were added to a small section of the video (see Fig. 2). The video is available here: https://osf.io/r9hm4/?view_only=c768a8f319ed4eb7bf89199a10bbd584

*Stimuli headlines*. We kept the same images as Experiment 1 for the stimuli headlines but adapted the text to include words from the polarisation

dictionary in the affectively-polarised (derogatory) stimuli[12] (see Supplementary Table 2).

*Measures*. We adapted the following measures from Experiment 1.

*Brexit identity strength*. Due to constraints on the length of the survey, based on factor analysis of the five item identity strength measure (see Supplementary Tables S11 and S12), the measure was shortened to three items.

*Sharing likelihood*. The four-point Likert scale employed in Experiment 1 was substituted for a 10 point linear scale based on work that suggests that using 10-point scales increases the variation in responding over 5 and 7 point scales[61,62]. Point 1 was labelled as "I definitely **would not** re-share this article on social media", and 10 labelled as "I definitely **would** re-share this article on social media".

*Clicking likelihood*. The four point Likert scale employed in experiment one was substituted for a 10 point linear scale, with 1 labelled as "I definitely **would not** click on this article to find out more", and 10 labelled as "I definitely **would** click on this article to find out more".

The following measures were added to the second experiment for the purposes of exploratory analysis, but no analyses involving these measures are reported in the current paper.

*Cognitive reflection test*. A three-item measure of reflective reasoning, used extensively across a broad set of literature to measure cognitive reasoning abilities[63,64]. Three problems are posed to respondents, each of which provokes an intuitive (and incorrect) response which they must resist to reach the correct answer. The measure is scored from 0 to 3, with each correct answer adding one to the total score. High scores on this measure are believed to reflect greater reflective reasoning[65].

*Need for chaos scale*. An eight-item measure of the desire to disrupt established political and social orders[66]. Responses are recorded on a seven-point Likert scale, ranging from strongly disagree (1), to strongly agree (7). An overall score is calculated by summing responses, with higher scores indicating a greater "need for chaos"[67].

All other measures from Experiment 1 were included unchanged.

## Experiment 3
**Participants**. Our target and pre-registered sample size was 1540. However, as the data collection was part of the Polarization Research Lab's Request for Proposals, the actual sample size exceeded this number, totalling 2000 participants recruited from a representative American sample via YouGov. In accordance with our pre-registered analysis plan, we excluded Independents ($N = 348$). The final analysis included 1652 participants (879 Women, 773 Men). Participants' age ranged from 18 to 90 ($M = 50.34$, $SD = 18.08$). See Supplementary Table S13 for sample characteristics.

**Design, materials & procedure**. The design and procedure followed the same design and procedure as in Experiments 1–2, with the exception that we did not measure polarisation after the intervention. Respondents were randomly assigned to either the inoculation condition or the

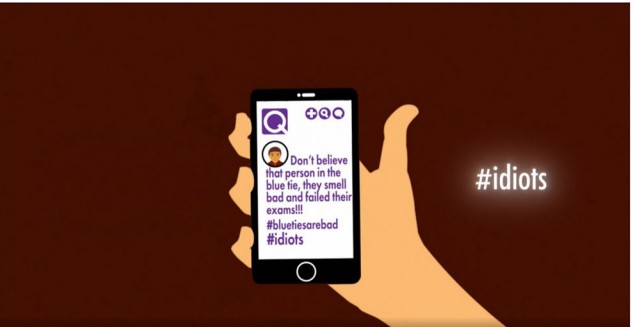
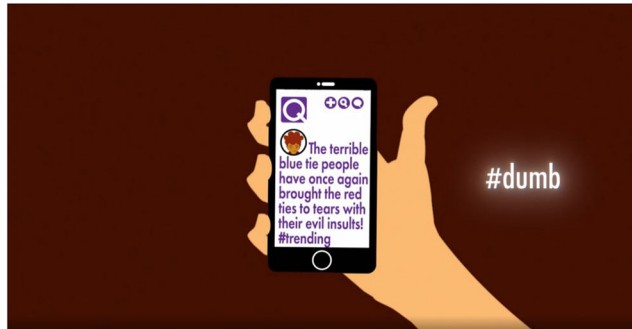

**Fig. 2 |** Snips from the revised inoculation video for Experiment 2, including examples of words included from the polarisation dictionary.

control condition. In the inoculation condition, participants were asked to read a short text explaining how some people benefit from spreading polarisation and were then given concrete examples of political headlines using affectively polarised words. In the control condition, participants read a short text about the US government system adapted from Wikipedia. Then, participants were shown six pro-attitudinal news items (three polarised based on the relevant issue, viz. Roe vs. Wade, and three that use affectively polarised language). After each item, participants were asked about their likelihood of engaging and sharing the news article.

The stimuli were curated in a data-driven fashion. We used an open dataset of **real** Roe vs. Wade tweets, manually labelled as pro-choice or pro-life by a commercial data labelling platform (https://www.surgehq.ai/blog/dataset-of-roe-v-wade-tweets-labeled-by-abortion-stance). We extracted tweets that contained news links and analyzed the headlines with the polarisation dictionary[12]. We selected from these **real published** headlines based on their polarisation score (issue vs. affective), and in some cases, inserted one affectively polarised word to increase its salience.

## Experiment 4

**Participants.** Our pre-registered (https://aspredicted.org/2x49-jmp8.pdf) target sample size was 690 participants. Based on a power analysis, we aimed to collect 620 participants to achieve 80% power, assuming a hypothesised effect size of $d = 0.2$ and an alpha level of 0.05. Anticipating that ~10% of the sample would be excluded, we adjusted our final target to 690 participants. Participants received £0.90 for their participation in the study.

In line with our pre-registered analysis plan, we excluded participants who completed the writing task unusually quickly. Specifically, we applied the lenient exclusion criterion recommended in previous work[68], which flags individuals whose completion time is 50% faster than the median for their respective group. This approach identifies those with abnormally fast response behaviours. Although we initially recruited 692 participants, applying this criterion reduced the final sample size to 599.

Our sample was designed to be representative of the US population across both demographic variables and political affiliation. The political breakdown of the final sample was as follows: 200 Democrats, 184 Republicans, 210 Independents, and 5 participants who preferred not to disclose their affiliation. The age of participants ranged from 18 to 85 years ($M = 46.79$, $SD = 15.93$).

**Design.** The design followed the same between-subjects design as in all previous experiments, with and inoculation group and a control group.

**Materials.** *Videos.* Inoculation video was the same as in Experiments 2 and 3. For the control condition, participants watched a short video about Congress (https://www.youtube.com/watch?v=irIuYLzj_Qc). After watching the video, participants were asked to describe in one sentence what the video was about, as a comprehension check.

*Measures.* Writing task. We asked participants to express their views on abortion and its legal status in the USA in at least 80 words.

Political affiliation. Participants were asked to indicate their political affiliation by selecting one of the following options: Democrat, Republican, Independent, or Prefer not to answer.

**Natural language processing.** We extracted average embeddings of affective polarisation using the polarisation dictionary from ref. 12. Next, we applied the Distributed Dictionary Representation (DDR) method[69] to the text written by the participants. This process involves tokenizing each text, extracting its embeddings using the FastText model[70], and averaging these embeddings. We then calculated the cosine similarity between each text and the averaged affective polarisation dictionary vector.

## Experiment 5

**Participants.** Similar to Experiment 4, our pre-registered (https://aspredicted.org/q7yw-zg4p.pdf) target sample size was 690 participants. In line with our pre-registered analysis plan, we excluded participants who completed the writing task unusually quickly. Due to a higher-than-expected exclusion rate in Experiment 4, we applied an even more lenient exclusion criterion, flagging individuals whose completion time was 50% faster than the 70th percentile based on time spent on the writing task. Additionally, we asked participants if they had used AI to complete the task, informing them that such use would affect their compensation (£0.90). We initially recruited 691 participants. Applying the speeding criterion resulted in the exclusion of 68 participants, while the AI criterion excluded an additional 22. The final sample size was 645 participants, with 326 identifying as pro-choice and 319 as pro-life (gathered from Prolific pre-screeners). The political breakdown of the sample was as follows: 223 Democrats, 233 Republicans, 177 Independents, and 12 participants who preferred not to disclose their affiliation. The age of participants ranged from 19 to 80 years ($M = 42.27$, $SD = 13.39$).

**Design.** The design followed the same between-subjects design as in all previous experiments, with and inoculation group and a control group.

**Materials.** *Measures* Writing task. Participants were asked to respond to a (fake) social media post that opposed their own stance on the issue. Pro-life participants were shown the following message:

"My body, my choice. If you think you can control what I do with my body, you're delusional. Stay out of my uterus. #ProChoice #ReproductiveRights"

Pro-choice participants, on the other hand, were presented with this message:

"Abortion is murder, plain and simple. You're ending a human life— no excuses. If you can't handle the consequences, don't have sex. #ProLife #EndAbortion"

Participants also completed the same political affiliation measure as in Experiment 4.

## Experiment 6

**Participants.** As in Experiments 4 and 5, our pre-registered (https://aspredicted.org/2scr-dcvv.pdf) target sample size was 690 participants. We applied the same exclusion criteria as in Experiment 5 and compensated participants £0.90 for their participation.

We initially recruited 689 participants. After applying the speeding criterion, 34 participants were excluded, and an additional 23 were excluded based on the AI usage criterion. This left a final sample size of 634 participants, with 322 identifying as pro-choice and 312 as pro-life (based on Prolific pre-screening). The political breakdown of the sample was as follows: 209 Democrats, 254 Republicans, 158 Independents, and 13 participants who preferred not to disclose their affiliation. The age of participants ranged from 18 to 95 years ($M = 42.04$, $SD = 13.31$).

**Design.** The design replicated the between-subjects structure used in previous experiments, featuring an inoculation group and a control group. This experiment specifically replicated Experiment 4, with participants pre-screened based on their stance (pro-life/pro-choice).

## Reporting summary

Further information on research design is available in the Nature Portfolio Reporting Summary linked to this article.

**Fig. 3 | Flow diagram of procedure for experiments one and two.**

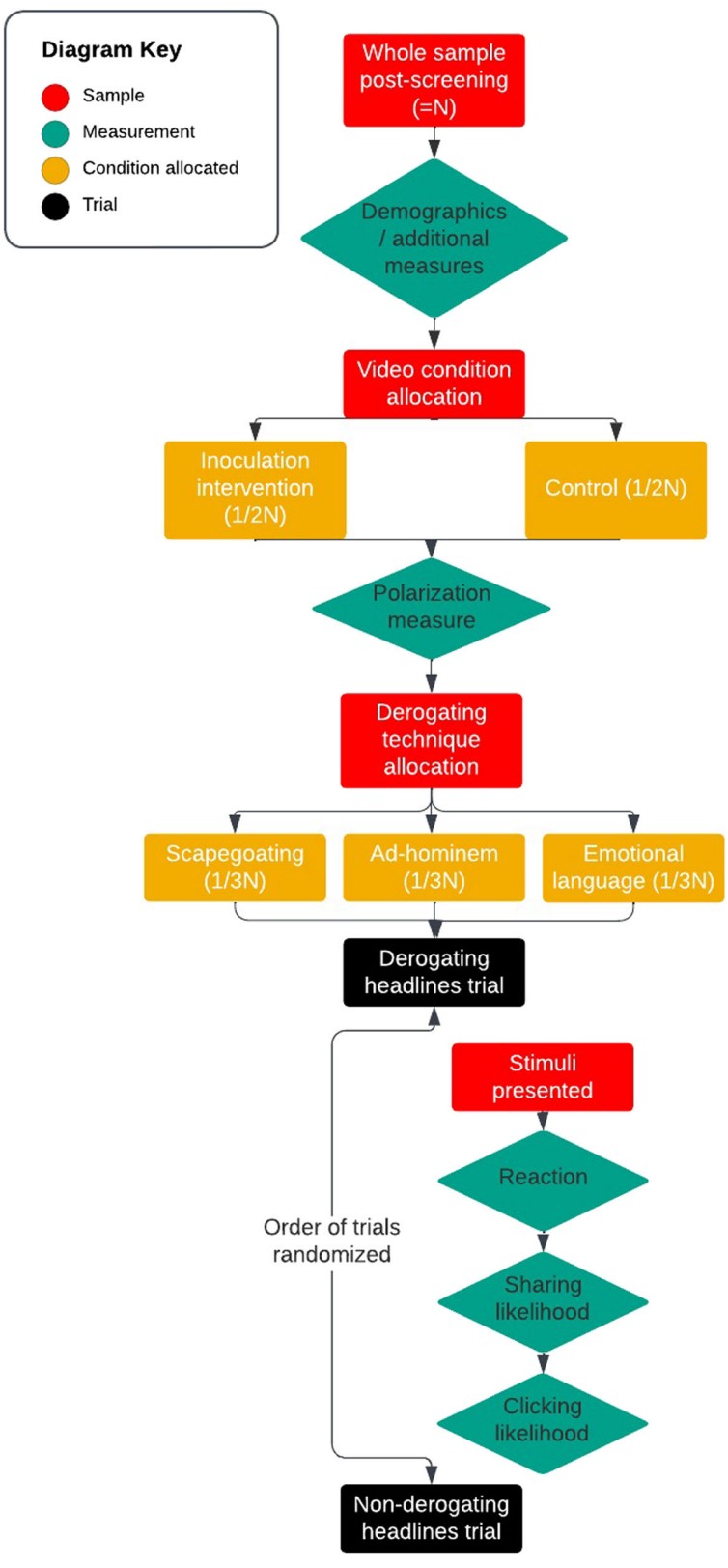

## Results

### Study set I: overview

Across Experiments 1–3, participants were either exposed to the inoculation intervention, or a control (manipulated between-subjects). In Experiments 1 (pre-registration: https://osf.io/zakvj) and 2, participants then completed a measure of their level of polarisation towards their respective out-group in the context of Brexit (in the UK). Test stimuli consisted of two types of headlines: one headline had affectively polarised (out-group derogating) content and the other was issue-polarised, i.e. it aligned with participants' political views but did not include features of affectively polarised content. In Experiment 3, respondents saw six different headlines, half affectively polarised, and half issue polarised. Respondents reported how likely they were to engage with each of the headlines if they encountered them on social media. In Experiments 1 and 2, we used three engagement measures:

**Fig. 4 | Results of Experiments 1 and 2: effects of video condition and polarisation on engagement likelihood.** The impact of video condition (inoculation vs. control) and polarisation (affective vs. issue) on engagement likelihood (z-scored) in Experiment 1 (**A**; $n = 760$) and Experiment 2 (**B**; $n = 864$). Bands denote 95% confidence intervals.

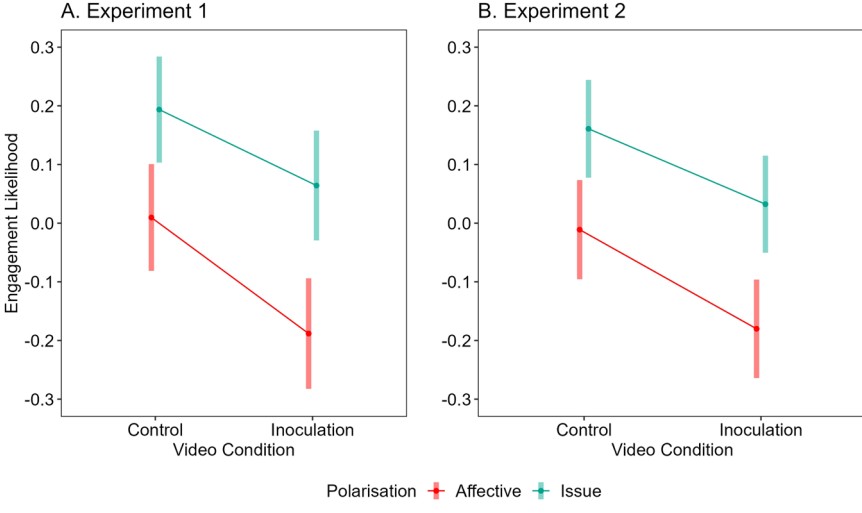

sharing likelihood, and clicking likelihood for the post (our pre-registered primary outcomes), and choice of emoji reactions (an exploratory measure, presented in a format mimicking Facebook's; see Fig. 3). In Experiment 3 (pre-registration: https://aspredicted.org/rxvs-qnjg.pdf; https://aspredicted.org/blind.php?x=61Q_PPX), we reduced the sharing and clicking likelihood to one item asking how likely participants were to like or share the post. For brevity, we report here the consolidation of sharing and clicking likelihood scores into a single engagement metric for all 3 experiments. This deviates from our pre-registered analyses for Experiment 1 to examine sharing and clicking likelihood separately; for the pre-registered separated analyses, see Supplementary Information. Previous experiments have found self-reported engagement (e.g. intention to share) correlated with actual sharing behaviour ($r = 0.44$) [71].

### Experiment 1
As expected, the inoculation was successful in reducing overall self-reported engagement, $F(1, 717) = 7.33$, $p < 0.01$, $\eta_G^2 = 0.01$, 90% two-tailed CI [0.00, 0.03] (see Fig. 4). See Supplementary Table S14 for descriptive statistics.

However, contrary to expectations, participants were significantly more likely to engage with *non-affectively polarised* than affectively polarised headlines, $F(1, 717) = 62.15$, $p < 0.001$, $\eta_G^2 = 0.02$, 90% two-tailed CI [0.01, 0.04]. This runs counter to recent findings that emphasise the role of outgroup-derogating (affectively polarised) content as a predictor of engagement with posts on social media [32,39].

The interaction effect we expected was also not observed. The ANOVA found no significant interaction between intervention condition (video seen) and headline type for either outcome, $F(1, 717) = 1.53$, $p = 0.213$, $\eta_G^2 < 0.001$, 90% two-tailed CI [0.00, 0.00], suggesting that the video reduced engagement with both affective and issue polarised stimuli. This was contrary to our hypothesis and could represent a problem for our inoculation, since issue-polarised content can still contribute to discourse characteristic of a healthy democracy, and therefore it is not desirable to reduce engagement with it [72]. Ideally, the inoculation should target only affectively polarised content, which worsens the health of democracy [16,73].

Experiment 1 therefore confirmed that the inoculation reduced engagement, but not just with affectively polarised stimuli as we had theorised. For Experiment 2, we altered our inoculation to take a more bottom-up approach, incorporating words into both the inoculation and the stimuli which were identified to often appear in affectively polarised social media posts [12].

### Experiment 2
We repeated the between-within ANOVA for engagement with the headlines (consolidated sharing and clicking likelihood; see Supplementary Tables S15 and S16 for separate analyses) in Experiment 2, which replicated the findings of Experiment 1, with a similar effect size. The inoculation significantly reduced engagement with affectively polarised political content, $F(1, 860) = 7.35$, $p < 0.01$, $\eta_G^2 = 0.01$, 90% two-tailed CI [0.00, 0.02]. As in Experiment 1, we also found that participants reported significantly greater engagement with issue polarised than affectively polarised content, $F(1, 860) = 59.57$, $p < 0.001$, $\eta_G^2 = 0.01$, 90% two-tailed CI [0.00, 0.02]. See Supplementary Table S17 for descriptive statistics.

To see if differences in cognitive reflection explained some of our observed effects, we ran an exploratory analysis, adding cognitive reflection score as a covariate. The effect of our inoculation was almost totally unchanged, $F(1, 857) = 6.99$, $p < 0.01$, $\eta_G^2 = 0.01$, 90% two-tailed CI [0.00, 0.02]. Cognitive reflection score was not a significant predictor of engagement, $F(3, 857) = 1.52$, $p = 0.21$, $\eta_G^2 = 0.00$, 90% two-tailed CI [0.00, 0.00].

As in Experiment 1, there was no significant interaction effect between intervention condition and stimuli type $F(1, 860) = 0.66$, $p = 0.427$, $\eta_G^2 = 0.001$, 90% two-tailed CI [0.00, 0.01]. This indicates that the intervention not only reduced engagement with headlines featuring affectively polarised language, but also those on different sides of the political issue without such language (see Fig. 4).

### Experiment 3
Experiment 3 extended Experiments 1–2 in three major ways: (1) our sample and polarised issue pertained to the American context (Roe vs. Wade —an issue that was highly emotionally charged at the time of data collection); (2) instead of synthetic stimuli, we used real headlines shared on social media — refer to method for more info, and; (3) we shifted from video-based inoculation to a text-based inoculation. This experiment was selected in a Request for Proposals by the Polarization Research Lab[74] and was fully pre-registered, see: https://aspredicted.org/blind.php?x=61Q_PPX. As before, we wanted to investigate if inoculating against the use of affectively polarised language affects people's willingness to interact with news headlines involving partisan conflict.

We hypothesized that (a) the inoculation would be effective and (b) the intervention would be more effective for affectively polarised items than issue-based items.

We conducted a 2 (inoculation/control; between-subjects) ×2 (affective/issue polarisation; within-subjects) ANOVA. In line with our predictions, there was a main inoculation effect, such that the intervention reduced the general tendency to share polarised news, $F(1, 1650) = 56.45$, $p < 0.001$, $\eta_G^2 = 0.03$, 90% two-tailed CI [0.02, 0.04], a main effect for polarisation type, $F(1, 1650) = 7.97$, $p = 0.005$, $\eta_G^2 < 0.01$, 90% two-tailed CI [0.00, 0.01], and importantly a significant interaction such that the intervention was especially effective in reducing the likelihood of engaging with affectively

**Fig. 5 | Engagement likelihood for experiment three ($n = 1652$) by inoculation and polarisation type.** Inoculation is a between-subjects factor; polarisation is a within-subjects factor. Bands denote 95% confidence intervals.

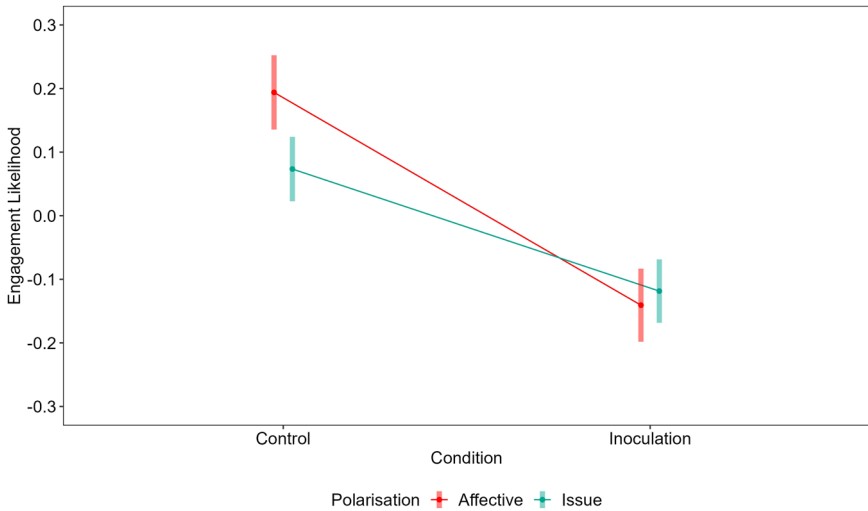

polarised news items $F(1, 1650) = 16.78$, $p < 0.001$, $\eta_G^2 = <0.01$, 90% two-tailed CI [0.00, 0.01] (see Fig. 5).

For completeness, we also modelled our main analysis in Experiments 1–3 as a series of linear mixed models with a random intercept by participant. We started with a null model (intercept only) and continued to add predictors until it comprised two main effects and their interaction. See Supplementary Tables S18–S20 for results and model comparison via BF Inclusion.

### Study set II: overview

Similar to Study Set I, participants in the three experiments were either exposed to the inoculation intervention or assigned to a control group (manipulated between subjects). In Experiment 4, after watching the video, participants were asked to write a brief paragraph expressing their views on abortion. We then conducted a text analysis to assess the degree of affective polarisation present in their responses, utilising the DDR[69] method and a validated polarisation dictionary[12]. For further methodological details, refer to the Method section.

In Experiment 5, participants were pre-screened for their stance on abortion (pro-life or pro-choice) using Prolific's screening tools. After viewing the video, they were exposed to counter-attitudinal messages resembling social media posts and were asked to comment on the post, sharing their own views.

**Table 1 | Experiment 4: Regression results predicting affective polarisation**

| Predictors | Estimates | CI | p |
|---|---|---|---|
| (Intercept) | 0.54 | 0.54–0.55 | <0.001 |
| Group [Inoculation] | −0.00 | −0.01–0.00 | 0.357 |
| Political [Independent] | −0.01 | −0.01–0.00 | 0.070 |
| Political [Prefer not to answer] | 0.02 | −0.01–0.06 | 0.153 |
| Political [Republican] | −0.00 | −0.01–0.01 | 0.793 |
| Group [Inoculation] × Political [Independent] | 0.01 | −0.00–0.02 | 0.122 |
| Group [Inoculation] × Political [Prefer not to answer] | −0.01 | −0.05–0.03 | 0.677 |
| Group [Inoculation] × Political [Republican] | −0.00 | −0.01–0.01 | 0.886 |
| Observations | 599 | | |
| $R^2$/Adjusted $R^2$ | 0.014/0.002 | | |

Experiment 6 integrated both approaches: participants were pre-screened for their stance, but this time they were asked to share their views without being shown any counter-attitudinal messages.

We hypothesised that inoculating against affective polarisation would reduce the amount of affectively polarised text participants wrote compared to the control.

### Experiment 4

We fitted a regression model to predict affective polarisation, measured by the cosine similarity between participants' text and the affective component of the polarisation dictionary. The model included intervention type (inoculation/control), political affiliation, and their interaction as predictors. None of the key effects—either the inoculation intervention or its interactions—reached statistical significance (see Table 1). When compared against a model that does not include the intervention, the more parsimonious model incorporating only political affiliation is decisively favoured ($BF_{10} < 0.0001$).

### Experiment 5

We fitted a regression model to predict affective polarisation, measured by the cosine similarity between participants' text and the affective component of the polarisation dictionary. The model included intervention type (inoculation/control), political affiliation, stance (pro-life/pro-choice) and their interaction as predictors. Again, none of the key effects—either the inoculation intervention or its interactions—reached statistical significance (see Table 2). When compared against a model that does not include the intervention, the more parsimonious model incorporating political affiliation and stance is decisively favoured ($BF_{10} < 0.0001$).

### Experiment 6

We fitted a regression model to predict affective polarisation, measured by the cosine similarity between participants' text and the affective component of the polarisation dictionary. The model included intervention type (inoculation/control), political affiliation, stance (pro-life/pro-choice) and their interaction as predictors. Again, none of the key effects—either the inoculation intervention or its interactions—reached statistical significance (see Table 3). When compared against a model that does not include the intervention, the more parsimonious model incorporating political affiliation and stance is decisively favoured ($BF_{10} < 0.0001$).

### Discussion

Considering Experiments 1–3 together, our findings show that a technique-based inoculation designed to undermine people's partisan motivations to engage with affectively polarised content could reduce engagement, in the

**Table 2 | Experiment 5: Regression results predicting affective polarisation**

| Predictors | Estimates | CI | p |
|---|---|---|---|
| (Intercept) | 0.54 | 0.54–0.55 | <0.001 |
| Group [Inoculation] | 0.00 | −0.00–0.01 | 0.286 |
| Stance [Pro-life] | 0.00 | −0.01–0.01 | 0.719 |
| Political [Independent] | −0.00 | −0.01–0.01 | 0.950 |
| Political [Prefer not to answer] | 0.02 | −0.01–0.04 | 0.231 |
| Political [Republican] | 0.01 | −0.00–0.02 | 0.222 |
| Group [Inoculation] × Stance [Pro-life] | −0.01 | −0.02–0.01 | 0.319 |
| Group [Inoculation] × Political [Independent] | −0.01 | −0.02–0.01 | 0.306 |
| Group [Inoculation] × Political [Prefer not to answer] | −0.04 | −0.09–0.00 | 0.051 |
| Group [Inoculation] × Political [Republican] | −0.01 | −0.03–0.01 | 0.175 |
| Stance [Pro-life] × Political [Independent] | −0.01 | −0.02–0.01 | 0.355 |
| Stance [Pro-life] × Political [Prefer not to answer] | −0.03 | −0.06–0.01 | 0.133 |
| Stance [Pro-life] × Political [Republican] | −0.01 | −0.02–0.01 | 0.297 |
| (Group [Inoculation] × Stance [Pro-life]) × Political [Independent] | 0.01 | −0.01–0.03 | 0.338 |
| (Group [Inoculation] × Stance [Pro-life]) × Political [Prefer not to answer] | 0.05 | −0.02–0.12 | 0.137 |
| (Group [Inoculation] × Stance [Pro-life]) × Political [Republican] | 0.02 | −0.01–0.04 | 0.181 |
| Observations | 645 | | |
| $R^2$/Adjusted $R^2$ | 0.019/−0.004 | | |

**Table 3 | Experiment 6: Regression results predicting affective polarisation**

| Predictors | Estimates | CI | p |
|---|---|---|---|
| (Intercept) | 0.54 | 0.53–0.54 | <0.001 |
| Group [Inoculation] | 0.00 | −0.01–0.01 | 0.620 |
| Stance [Pro-life] | −0.00 | −0.01–0.01 | 0.765 |
| Political [Independent] | −0.00 | −0.01–0.01 | 0.450 |
| Political [Prefer not to answer] | 0.02 | −0.01–0.05 | 0.115 |
| Political [Republican] | −0.01 | −0.02–0.00 | 0.217 |
| Group [Inoculation] × Stance [Pro-life] | 0.01 | −0.01–0.02 | 0.448 |
| Group [Inoculation] × Political [Independent] | −0.00 | −0.01–0.01 | 0.756 |
| Group [Inoculation] × Political [Prefer not to answer] | −0.02 | −0.06–0.03 | 0.436 |
| Group [Inoculation] × Political [Republican] | 0.00 | −0.01–0.02 | 0.635 |
| Stance [Pro-life] × Political [Independent] | 0.01 | −0.01–0.02 | 0.383 |
| Stance [Pro-life] × Political [Prefer not to answer] | 0.00 | −0.03–0.04 | 0.812 |
| Stance [Pro-life] × Political [Republican] | 0.01 | −0.01–0.02 | 0.302 |
| (Group [Inoculation] × Stance [Pro-life]) × Political [Independent] | −0.02 | −0.04–0.01 | 0.168 |
| (Group [Inoculation] × Stance [Pro-life]) × Political [Prefer not to answer] | −0.02 | −0.07–0.04 | 0.590 |
| (Group [Inoculation] × Stance [Pro-life]) × Political [Republican] | −0.01 | −0.03–0.01 | 0.289 |
| Observations | 633 | | |
| $R^2$/Adjusted $R^2$ | 0.024/−0.000 | | |

form of re-sharing and clicking, with such content on social media. Previous inoculation studies employing survey-based methodologies found small but significant effects of technique-based inoculation upon identification sharing of misleading content[49,52]. Experiments 1–3 extend these findings to technique-based inoculation for reducing intentions to share partisan content. Though our effect sizes are small, given the frequent engagement affectively polarised content receives on social media–often millions of times–the results nevertheless imply that if the effects were translated into real-world behaviour, the intervention could have a tangible impact in slowing and reducing the spread of partisan misinformation[48,75].

Two intriguing differences exist between results from the UK and US samples. First, in the UK, affectively polarised stimuli received significantly less engagement than issue-polarised stimuli, whilst in the US the reverse was true. Second, a significant interaction effect between the type of stimuli and intervention condition was observed in the US, but the same interaction was not significant in the UK. We can only speculate as to what underlies these differences.

Though both the political contexts for the UK and US portions of our studies are highly affectively polarised debates[56,59], Google Trends data suggests that interest in Brexit has overall decreased over the last five years, becoming less salient, whilst the opposite is true in the US when it comes to the abortion debate[76,77]. The controversial Dobbs vs. Jackson Women's Health Organisation ruling on abortion concluded in the year prior to Experiment 3 in the US and affected four in ten American voters' decisions on whether or not to vote in the US midterms[55] and is likely to be a key issue at the next presidential election in 2024. In contrast, the UK Brexit referendum concluded six years prior to Experiment 1. Furthermore, recent public opinion polling suggests that the opinions of leave voters on Brexit have shifted significantly since the vote, with only 13 in 20 believing Britain was right to leave the EU in recent polling compared to 19 in 20 in the months following the vote[78].

Social identity theory suggests that the greater the salience of political identities, the greater the desire to preserve the in-group's status and attack the out-group[5]. The greater salience of the issue in Experiment 3 may mean that partisans in the US sample had stronger feelings about the debate and were more accustomed to seeing partisan coverage of the issue than the UK samples in Experiments 1 and 2, for whom the salience of the Brexit debate may have been lower and awareness of the negative impacts of the polarised dialogue around Brexit higher. Therefore, the US sample may have had stronger partisan motivation and perceived higher political utility to engage with affectively polarised Roe vs. Wade than the UK sample for Brexit content.

In Experiments 4–6, we probed the effectiveness of the inoculation intervention further in the US context by extending it to a behavioural outcome measure: producing text when engaging in a potentially polarising debate. To our knowledge, ours is the first study to attempt inoculation against sharing/clicking on affectively polarising content, as well as the use of affectively polarised language when **generating** content. Overall, Experiments 4–6, which investigated the impact of our inoculation on the use of affectively polarised language in user generated text, found it has no significant effect.

This is contrary to our expectations, based on Experiments 1–3, which demonstrated a significant effect of our inoculation in reducing self-reported likelihood of engaging with affectively polarised content, in the form of headlines shared on social media. What underlies this discrepancy in our findings remains unclear.

One explanation that we believe deserves further study is the content of the inoculation itself. In the inoculation, we refer to 'the hate pushers' as a malevolent force that seek to disrupt democracies with the polarising content that they generate. Whilst we do suggest an active role for the user in asking that they 'focus on our similarities not on our differences', it is possible that the inoculation is not explicit enough in suggesting that the

user/viewer could themselves play an active role in generating hateful and affectively polarised content, and therefore themselves become a 'hate pusher'. Instead, the inoculation is focussed on the content generated by others. Therefore, we suggest that it is possible an inoculation that both highlights the risks of engaging with affectively polarised content generated by others and suggests a more active role for users in moderating the content they generate themselves could be more effective in reducing the use of affectively polarised language in user-generated content. This aligns with previous research showing that technique-focused inoculation interventions, while effective for technique detection, need to be supplemented with additional elements to transfer to related tasks like accuracy judgments[79].

It, therefore, remains an open question whether inoculation effects on intentions to engage with affectively polarising content would generalise to social media users' production of text on those platforms.

Further research is therefore needed to determine whether the robust effects found on the commonly-used intention-to-engage measures in previous literature[49,52] do indeed extend beyond these to real behaviours on social media platforms, or if text production is a specific case that is resistant to inoculation attempts. Given that people's language use and the type of content they consume is strongly associated with their personalities[80–82], it may be difficult for a single inoculation intervention to generate change in this measure.

## Limitations

Our findings are strengthened by the stringent selection criteria we applied in our sampling. In Experiments 1–3, we drew our samples from broader nationally and politically representative samples of the respective populations. Across all experiments, we ensured that only partisans were included in our final analyses (only those identifying as a "Leaver" or "Remainer" in the UK samples; identifying as a Republican nor a Democrat in Experiment 3, or as being pro-life or pro-choice in Experiments 4–6).

However, as with all studies that rely on a cross-sectional survey-based design, we are limited in our ability to determine the longevity of our inoculations effect or prove its ecological validity conclusively.

A fundamental limitation concerns the distinct types of engagement across our studies. While Experiments 1–3 examined low-effort engagement (clicking, sharing), Experiments 4–6 investigated content creation through writing tasks. These represent fundamentally different cognitive processes, which may explain why our intervention was effective only for simpler forms of engagement. Additionally, our writing tasks did not fully capture the interactive nature of social media environments, with participants potentially moderating their language due to researcher oversight and the absence of a real online audience.

Notwithstanding this limitation, previous findings on the duration of inoculation effects give cause for optimism. Though the endurance of the effect of inoculation is uncertain, some have found it can last two weeks[83,84]; others found it lasts up to three months. The means by which an inoculation of this kind would be rolled out online also make concerns about longevity of the effect less relevant. Regular top-up doses—perhaps triggered by encounters with posts containing lots of words that commonly appear in affectively polarised content—would serve to extend, and reinforce the inoculation's effect[85]. On the question of ecological validity, previous experiments employing similar designs found that self-reported engagement with stimuli was strongly correlated with real-world engagement on social media[27,71]. Furthermore, inoculation interventions that successfully improved discernment of fake news in the lab have had significant effects in large-scale field studies online[49].

A possible criticism of our inoculation is that it is designed to be politically symmetrical, and therefore could be seen to encourage equally the toleration of far-right views as well as the (typically more moderate) opposing views on the left. Previous literature has indicated that the spread of hyper-partisan content tends to be a politically asymmetrical problem, with those who produce and engage with affectively polarised disinformation typically being further to the political right[39,86–88]. This is exemplified by populist right-wing candidates and political movements that have benefitted from and in some cases actively endorsed and promoted affectively polarised content, such as Donald Trump in the US, Modi in India, and the Brexit campaign in the UK[89,90]. However, we believe our approach is in fact responsive to the asymmetrical slant in much of the contemporary polarisation. If it is primarily those on the far right who generate and propagate derogatory, affectively polarised content, our inoculations' focus on this content—and the explicit reference to "hate pushers" in our inoculation interventions—should reduce engagement with this group more than content supporting more moderate perspectives, much of which arguably is towards the centre and left of the political spectrum. If the debate were to feature affectively polarised content from both the extreme right and extreme left, we have reason to believe that the inoculation should work equally well on both[52].

Future research should focus on developing and testing interventions that can address both content consumption and creation in more naturalistic social media settings. Such work will be crucial for understanding how inoculation effects might translate to real-world behaviour across different forms of social media engagement.

## Conclusion

The current study confirms the efficacy of an inoculation intervention in the form of a short video, or a short piece of text, at reducing self-reported engagement with affectively polarised content online via clicking or sharing, but not producing affectively polarised text. While inoculation could be a scalable and effective tool to stifle the spread of hyper-partisan content on social media by undermining partisan motivations to engage with it, it may not be sufficient to reduce the generation of partisan content. Future research should confirm the inoculation's effectiveness in the context of reducing engagement with affectively polarised content in other polarised debates and whether inoculation against techniques that promote affective polarisation can have any effect on how individuals generate content for social media that relates to polarised debates.

## Data availability

All data used in this article are publicly available at https://zenodo.org/records/13996693 Data are also available on GitHub, here: https://github.com/fintans-123/Inoculation_polarisation.

## Code availability

All source code used in this article is publicly available at https://zenodo.org/records/13995889 Code is also available on GitHub, here: https://github.com/fintans-123/Inoculation_polarisation.

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

## Acknowledgements

DH and SL are supported by Horizon 2020 grant 964728 (JITSUVAX). AS and SL are supported by the Volkswagen Foundation (grant "Reclaiming individual autonomy and democratic discourse online: How to rebalance human and algorithmic decision making"). SL also acknowledges funding from the European Research Council (ERC Advanced Grant 101020961 PRODEMINFO), the Humboldt Foundation through a research award, and the European Commission (Horizon 2020 grant 101094752 SoMe4Dem). SL also receives funding from UK Research and Innovation through EU Horizon replacement funding grant number 10049415. The funders had no role in study design, data collection and analysis, decision to publish or preparation of the manuscript We acknowledge funding from The Polarization Research Lab's first RFP, which contributed the participants for Experiment 3. The Polarization Research Lab had no role in the analysis, decision to publish or preparation of the manuscript. The authors would like to thank the political team at YouGov UK, especially Adam McDonnell, for their help with the data collection for this study. Thanks also to Léo Bournas Milazzo and George Copsey, who produced the animation for the inoculation video. The authors would also like to thank Mattan S. Ben-Shachar.

## Author contributions

F.S.: Conceptualization, Data curation, Formal analysis, Visualization, Writing - original draft, Writing - review & editing; A.S.: Conceptualization, Data curation, Formal analysis, Supervision, Visualization, Writing - original draft, Writing - review & editing; D.H.: Conceptualization, Formal analysis, Supervision, Visualization, Writing - original draft, Writing - review & editing;

S.L.: Conceptualization, Supervision, Writing - original draft, Writing - review & editing.

## Competing interests

YouGov UK undertook the fieldwork for Experiments 1 and 2 of this project free of charge. Fintan Smith is a former employee of YouGov. YouGov had no input on the formulation, design or any other specifics of the experiment, and does in no way endorse any of the views expressed in this paper. Other authors declare no competing interests.
