## [Transparent Peer Review file · Communications Psychology]

Inoculation reduces social media engagement with affectively polarized content in the UK and US

Corresponding Author: Dr Almog Simchon

Version 0:

Decision Letter:

Dear Dr Simchon,

Thank you for your patience during the peer-review process. Your manuscript titled "Reducing social media engagement with affectively polarised content via inoculation" has now been seen by 3 reviewers, whose comments are appended below. You will see that they find your work of some potential interest. However, they have raised quite substantial concerns that must be addressed. In light of these comments, we cannot accept the manuscript for publication, but would be interested in considering a revised version that fully addresses these serious concerns.

We hope you will find the Reviewers' comments useful as you decide how to proceed. Should additional work allow you to address these criticisms, we would be happy to look at a substantially revised manuscript. If you choose to take up this option, please highlight all changes in the manuscript text file, and provide a detailed point-by-point reply to the reviewers.

Editorially, we consider it important that the revised manuscript provides additional empirical evidence to rule out the alternative explanations and which should ideally support the link between intentions and behaviour.

For existing and additional experiments, please transparently report all preregistered hypotheses, indicate which analyses are exploratory, and clearly flag any deviations from the preregistration. All preregistered analyses must be included, unless later identified as scientifically flawed or unfeasible. Please make study materials available for the reviewers.

I am attaching a checklist that details critical reporting requirements for the revised manuscript. Please attend to each item and ensure your manuscript is fully compliant. We are requesting that your manuscript aligns with these requirements as this facilitates the evaluation of your manuscript, reducing delays in re-review and potential future acceptance. If your revised manuscript is not aligned with these requests on major issues, such as those concerning statistics, it may be returned to you for further revisions without re-review. Additional information can be found in our style and formatting guide Communications Psychology formatting guide.

If the revision process takes significantly longer than five months, we will be happy to reconsider your paper at a later date, provided it still presents a significant contribution to the literature at that stage.

Please use the following link to submit your
- revised manuscript,
- point-by-point response to the referees' comments,

- cover letter (as a separate document),
- the Editorial Policy Checklist (see below),
- the Reporting Summary (see below), and
- the completed Editorial Request Table (attached):

Link Redacted

Thank you for the opportunity to review your work.

Best regards,

Jennifer Bellingtier

Jennifer Bellingtier, PhD
Senior Editor
Communications Psychology

REVIEWER EXPERTISE:

Reviewer #1 polarization

Reviewer #2 polarization

Reviewer #3 polarization

REVIEWER REPORTS:

Reviewer #1 (Remarks to the Author):

This manuscript demonstrates the effectiveness of an "inoculation" on people's willingness to engage with what the authors term "hyper-partisan" content. The intervention aims to raise awareness of the features and intentions of polarising content on social media. By making individuals aware of these features and intentions, the inoculation seeks to render this content less effective. The authors experimentally test the effectiveness of different versions of the intervention in the US and UK, employing various stimuli. The results provide support for the hypotheses. I believe that the paper will be of interest to many scholars in fields such as political science, psychology, and communication studies. There are also a couple of issues that are tackled less convincingly which I will outline below.

My first concern pertains to the mechanism, or the aspect of the intervention that generates the treatment effect. Respondents are informed (via video or text-based intervention) that polarising content on social media is detrimental to democracies, after which they are asked to rate such content. Consequently, the observed effects might stem from respondents' willingness to "assist" researchers in achieving perceived goals. Alternatively, the intervention may highlight socially undesirable behaviour in the survey/experiment, leading respondents to avoid actions considered socially undesirable after viewing the video (just to comply with instructions). In essence, it remains a bit uncertain whether the content of the video or text itself fosters awareness or learning that generates the observed results. I believe that the authors could strengthen the paper by ruling out alternative explanations for the observed effects.

My second concern pertains to the role of the suggested intervention. The experiments demonstrate the intervention's effectiveness when participants are exposed to it. However, achieving similar results outside of an experimental setting would require social media users to encounter the treatment. While discussing or evaluating the implementation of the intervention is not the objective of the paper, I believe it would be beneficial to understand better how the intervention could be used in practice. For instance, clarification on whether the intervention is intended for presentation in high schools or if it is social media companies that would need to feature it, despite the potential negative impact on their revenue (they might not be interested in an intervention that reduces engagement). If social media companies would need to roll out the video based intervention, I am wondering why people would watch more than just 3-5 seconds before moving to other content. Again, I understand this is not the focus of the paper, but information on the intention in the introduction and conclusion might still help the reader to understand the reasoning of the authors.

I found it extremely beneficial to watch the video-based treatment material used in Experiment 1. Although it was mentioned that the video was revised for Experiment 2, I was unable to locate a link to it. I would suggest making this revised material available as well. I did not find the text used in the text-based intervention for Experiment 3. I apologise if I overlooked it, but it was not present in the Materials section related to Experiment 3, where the link to the video appears in the context of Experiment 1.

I have a couple of smaller issues that could be addressed or communicated more transparently:

I understand that some of the fieldwork was pre-registered. It appears that Experiments 1 and 3 were pre-registered, but not

Experiment 2. Additionally, there were some differences in the pre-registrations. I would appreciate it if these differences could be communicated more transparently. Also, in some instances, analyses deviate from the pre-registrations (e.g., the note on page 10). It could facilitate readers' understanding if the authors could clarify how the results presented in the main differ from those in the appendix (E1-E2) rather than solely referring to the appendix tables.

The paper appears to use a combination of pre-existing stimulus material and material generated or revised for the experiment. In certain instances, it appears that "affectively polarised" words were added to pre-existing material. While I understand that tables in the appendix display all treatment material, I would appreciate additional clarification regarding the fabrication of material for the experiment. Specifically, it would be helpful to see text or a table explaining when and where material was fabricated. Given that norms vary across disciplines, I find it important to note carefully when material was fabricated, the extent to which stimulus material included factually incorrect information, and whether a debriefing took place to make respondents aware of factually incorrect information or manipulations.

The authors could communicate a bit more clearly whether the focus of the paper and the intervention is on reducing engagement with content that is polarising (regardless of its factual accuracy) or specifically on content that is both polarising and factually incorrect.

Finally, I would appreciate it if the authors could demonstrate how they ensure that what they define as "issue polarised" or "affectively polarised" is indeed perceived in that manner by respondents as well.

The title of the paper could state that the fieldwork was conducted in the US and the UK.

Reviewer #2 (Remarks to the Author):

I have attached the review report as PDF (due to formatting advantages and possibilities to highlight aspects). I suggest major revisions, thinking that the manuscript is definitely suitable for Communications Psychology.

Reviewer #3 (Remarks to the Author):

1) Lines 57-58, you write that: "Creators of such content have varied objectives, but often seek to influence and reduce trust in democratic processes by increasing group polarisation". I understand that this might be true in some cases, but it's also possible (and I would argue more commonly) that the dissemination of misinformation is a by-product of the financial incentives of sharing sensationalist or partisan news.

2) Line 66: capitalize Twitter and Facebook.

3) I really like how you summarized the debate on nudges on the paragraphs between the lines 87-108.

4) Lines 141-144 "Osmundsen et al. (2021) focused on all types of political news sharing in their analysis, finding that out-group derogating content in news headlines did not just predict sharing of fake news by out-group derogating content, but also the sharing of news from any outlet." As someone who was not familiar with this work from Osmundsen, I did not understand the key takeaway based on the sentence. Did out-group derogating content predict sharing news from any outlet? How does a type of content predict sharing of all types of content? Some clarification might help here.

5) Line 397: You mention YouGov exclusion criteria for speeders. Please specify what criterion they use (e.g., how many SDs above/below mean?)

6) Throughout the paper, you only have one example for the headlines for "remainers" and "leavers" I'd like to see examples of the derogatory vs non-derogatory headlines as well. That would be suitable for the methods section, as you're describing the stimuli, or for a figure/table.

7) As I started reading the paper, I kept asking myself about how much the intention of sharing translates into real world sharing. I see that you addressed this question by the end of the discussion (lines 354-357). This information should be earlier in the paper, perhaps by the end of introduction, when you're outlining the studies. I would also consider appropriate to report the magnitude of the correlation (e.g., "previous experiments employing similar designs found that self-reported engagement with stimuli was strongly associated with real world engagement on social media, with correlations ranging from x to y").

8) The way the stimuli are categorized is confusing to me. It seems like different sections of the paper use different labels. You mention Derogatory VS non-derogatory, issue polarized VS affectively polarized, scapegoating VS ad-hominem VS emotional language headlines. In your results section, it's sometimes hard to figure out what analysis you're referring to exactly. Part of the confusion might be because some of these terms are used interchangeably, but I couldn't tell with certainty if that was the case. For instance, study 1 reports a) reduced overall sharing, b) participants engaged more with non-derogatory content, and c) the interaction between intervention type and headline type (issue polarized VS affectively

polarized) was not significant. In study 2, you say that you replicated the results of study 1. However, the first result you report is that “The inoculation significantly reduced engagement with affectively polarised political content”. What finding from study 1 is this replicating? a, b, or c? It seems like you’re referring to result (c), but it didn’t make sense because study 1 did not find that it was significant.

Sometimes you refer to your conditions as derogatory VS non-derogatory and sometimes as affectively polarized VS issue polarized. If they are used interchangeably, please choose only one, as two names for the same categorization make readers confused. I think derogatory and non-derogatory are more intuitive terms.

9) In lines 189 – 193, you describe the strategies for inoculation as top-down/bottom-up. These strategy names could more intuitively be called rationality-based and emotion-based, or something similar. Top-down and bottom-up might make sense for the readers from the brain sciences, but the audience for this paper seems to be broader, so I’d avoid that.

10) Throughout your discussion section, you mention a few times that your intervention was successful to decrease the amount of affectively polarized content. However, that is unequivocally true only for experiment 3. In studies 1 and 2, the intervention decreased all types of sharing, which is not the same as saying that your intervention worked as hypothesized. Please explore in your discussion what could be some of the negative consequences of such intervention decreasing overall sharing behavior. What could happen if people decreased sharing of non-polarized stimuli? Is that something desirable?

11) Lines 354-357: “On the question of ecological validity, previous experiments employing similar designs found that self-reported engagement with stimuli was strongly correlated with real world engagement on social media (Guess et al., 2019; Mosleh, Pennycook, & Rand, 2020).” Throughout the paper, the question about ecological validity was popping in my mind, if it was stated earlier in the paper, I might have dismissed some skepticism earlier. This information should be earlier in the paper, perhaps by the end of introduction or beginning of results, when you’re outlining the study. I would also consider appropriate to report the magnitude of the correlation (e.g., “previous experiments employing similar designs found that self-reported engagement with stimuli was strongly associated with real world engagement on social media, with correlations ranging from x to y”).

12) Minor note: First sentence of abstract feels a little out of place because the focus is on the agent “malevolent actors” rather than the action “disruption of democracies”. This is a matter of style, but I would rewrite it to avoid too much attention to the malevolent actors. (On a first read it sounds a little sensationalist).

EDITORIAL POLICIES

We ask that you ensure your manuscript complies with our editorial policies and reporting requirements.

To that end, we require revised manuscripts to be accompanied by two completed items: a reporting summary that collects information on study design and procedure, and an editorial policy checklist that verifies compliance with all required editorial policies

- <https://www.nature.com/documents/nr-reporting-summary.zip>>Nature Research Reporting Summary
- <https://www.nature.com/documents/nr-editorial-policy-checklist.pdf>>Editorial Policy Checklist

All points on the policy checklist must be addressed. Your revised manuscript can only be sent back to the referees if these checklists are completed and uploaded with the revision.

Notes: If you have submitted a Stage 1 Registered Report, Review, Primer, Comment, or Perspective you do not need to submit these forms. If you have already submitted these forms, you may disregard this request.

** Visit Nature Research's author and referees' website at <http://www.nature.com/authors> for information about policies, services and author

benefits**

Communications Psychology is committed to improving transparency in authorship. As part of our efforts in this direction, we are now requesting that all authors identified as 'corresponding author' create and link their Open Researcher and Contributor Identifier (ORCID) with their account on the Manuscript Tracking System prior to acceptance. ORCID helps the scientific community achieve unambiguous attribution of all scholarly contributions. You can create and link your ORCID from the home page of the Manuscript Tracking System by clicking on 'Modify my Springer Nature account' and following the instructions in the link below. Please also inform all co-authors that they can add their ORCID to their accounts and that they must do so prior to acceptance.

Version 1:

Decision Letter:

Dear Dr Simchon,

Your manuscript titled "Reducing social media engagement with affectively polarised content via inoculation: Six experiments in the UK and US" has now been seen by our reviewers, whose comments appear below. In light of their advice I am delighted to say that we are happy, in principle, to publish a suitably revised version in Communications Psychology.

We therefore invite you to revise your paper one last time to address the remaining concerns of our reviewers and a list of editorial requests. At the same time we ask that you edit your manuscript to comply with our format requirements and to maximise the accessibility and therefore the impact of your work.

EDITORIAL REQUESTS:

We appreciate the additional studies 4-6, however the standards of the journal require that you provide positive evidence for the null findings, as otherwise we do not allow for them to be interpreted. We recommend adding equivalence tests and request you discuss the limitations of the new studies in the Limitations section.

SUBMISSION INFORMATION:

OPEN ACCESS:

* CODE AVAILABILITY: All Communications Psychology manuscripts must include a section titled "Code Availability" at the end of the methods section. We require that the custom analysis code supporting your conclusions is made available in a publicly accessible repository at this stage; please choose a repository that generates a digital object identifier (DOI) for the code; the link to the repository and the DOI must be included in the Code Availability statement. Publication as Supplementary Information will not suffice.

* DATA AVAILABILITY:

Link Redacted

Best regards,

Jennifer Bellingtier

Jennifer Bellingtier, PhD
Senior Editor
Communications Psychology

REVIEWERS' EXPERTISE:

Reviewer #2 polarization
Reviewer #3 polarization

REVIEWERS' COMMENTS:

Reviewer #2 (Remarks to the Author):

I thank the reviewers for addressing all of my comments. Given these clarifications and content-additions and the immense empirical additions in the revised manuscript with clearer results that provide a more nuanced understanding of the limits of the inoculation effect (i.e., on actual behavior), I suggest to accept the manuscript.

Reviewer #3 (Remarks to the Author):

I appreciate that the authors have addressed all my previous comments in the revised manuscript.

The addition of three new studies to respond to other reviewers' feedback is noted, and I commend the authors for employing the DDR approach to analyze the new data.

However, I have concerns regarding the implementation of the writing task in Studies 4 and 6, especially in relation to addressing the feedback about measuring real engagement. The writing task, while informative, does not seem to directly tackle the issue of genuine engagement with social media interactions. This task does not mimic the interactive nature of social media environments where participants engage with others' content. Additionally, the behavior assessed in the writing task appears tangential to the main independent variable and the broader discussion within the manuscript. If the intervention focuses on how individuals react to online stimuli, it is unclear why producing a written text on a political topic, such as personal opinions on abortion, would be influenced by the experimental manipulation.

Study 5 presents a task more aligned with the dependent variable of interest—participants' reactions to content they might encounter online, an online post. However, it remains unclear how this task better captures real engagement compared to the measures used in Studies 1-3, as it is not capturing the dynamics of online interaction. Furthermore, several potential confounding factors could have influenced participants' responses. For instance, participants might temper their emotional reactions knowing that: (a) their responses are being reviewed by researchers, and (b) their written input will not be publicly posted or read by the original comment's author or an online audience.

Finally, I believe there are fundamental differences between the cognitive demands of writing a response and the simpler

actions of liking or sharing content online. The last three studies appear to shift between the concepts of content consumption and content creation. The effort required to write a detailed opinion is significantly greater than the cognitive load associated with engaging through likes or shares. Even if the inoculation intervention proves effective for these more passive forms of engagement, I do not think it is fair to assume that it would generalize to content creation. While these behaviors may be interconnected, the rationale for using content creation as a dependent variable is unclear, given the paper's primary focus on content consumption and engagement.

In conclusion, while I appreciate the authors' thorough work, I find the manuscript to have been more cohesive with the initial three studies only. I leave the decision to your discretion but encourage careful consideration of the alignment between the research questions and the dependent variables assessed in the later studies.

REVIEWER REPORTS:

Reviewer #1 (Remarks to the Author):

This manuscript demonstrates the effectiveness of an "inoculation" on people's willingness to engage with what the authors term "hyper-partisan" content. The intervention aims to raise awareness of the features and intentions of polarising content on social media. By making individuals aware of these features and intentions, the inoculation seeks to render this content less effective. The authors experimentally test the effectiveness of different versions of the intervention in the US and UK, employing various stimuli. The results provide support for the hypotheses. I believe that the paper will be of interest to many scholars in fields such as political science, psychology, and communication studies. There are also a couple of issues that are tackled less convincingly which I will outline below.

RESPONSE:

Thank you for these comments and suggestions to improve the paper. We respond to the issues raised point by point below, with reference to where we have revised the paper.

My first concern pertains to the mechanism, or the aspect of the intervention that generates the treatment effect. Respondents are informed (via video or text-based intervention) that polarising content on social media is detrimental to democracies, after which they are asked to rate such content. Consequently, the observed effects might stem from respondents' willingness to "assist" researchers in achieving perceived goals. Alternatively, the intervention may highlight socially undesirable behaviour in the survey/experiment, leading respondents to avoid actions considered socially undesirable after viewing the video (just to comply with instructions). In essence, it remains a bit uncertain whether the content of the video or text itself fosters awareness or learning that generates the observed results. I believe that the authors could strengthen the paper by ruling out alternative explanations for the observed effects.

RESPONSE:

This is an interesting question that pertains more generally to whether inoculation interventions would still be effective in the field. We used stimuli and response measures similar to previous work in this domain that showed such ratings do translate to actual behaviour in field tests outside of a lab setting. This suggests that inoculation effects such as ours cannot simply be explained away by social desirability or demand effects.

Nonetheless, we agree that it is important to assess the limits of the intervention, particularly knowing that intentions do not always translate to behaviour. We thus conducted three additional experiments with the objective of gathering empirical data showing the effects of the inoculation intervention on people's linguistic behaviour- the production of free written text - as a different outcome measure, and one that went beyond assessing intentions. Our additional studies did not yield the expected results, but they provide important findings on the limitations of inoculation: while it was effective based on test items that are commonly used, the generalisability of the effect to other contexts such as written text production remains uncertain.

My second concern pertains to the role of the suggested intervention. The experiments demonstrate the intervention's effectiveness when participants are exposed to it. However, achieving similar results outside of an experimental setting would require social media users to encounter the treatment. While discussing or evaluating the implementation of the intervention is not the objective of the paper, I believe it would be beneficial to understand better how the intervention could be used in practice. For instance, clarification on whether the intervention is intended for presentation in high schools or if it is social media companies that would need to feature it, despite the potential negative impact on their revenue (they might not be interested in an intervention that reduces engagement). If social media companies would need to roll out the video based intervention, I am wondering why people would watch more than just 3-5 seconds before moving to other content. Again, I understand this is not the focus of the paper, but information on the intention in the introduction and conclusion might still help the reader to understand the reasoning of the authors.

RESPONSE:

The reviewer rightly highlights that for an intervention to be effective beyond the lab, they would need to encounter the video. Field studies on YouTube have shown that this is indeed viable (Roozenbeek et al., 2022; <https://doi.org/10.1126/sciadv.abo6254>). We have now included this in the introduction:

For example, Roozenbeek et al. (2022) demonstrated that an inoculation video could be implemented effectively within the YouTube platform to achieve a highly scalable effect.

I found it extremely beneficial to watch the video-based treatment material used in Experiment 1. Although it was mentioned that the video was revised for Experiment 2, I was unable to locate a link to it. I would suggest making this revised material available as well. I did not find the text used in the text-based intervention for Experiment 3. I

apologise if I overlooked it, but it was not present in the Materials section related to Experiment 3, where the link to the video appears in the context of Experiment 1.

RESPONSE:

The video for experiment two has been uploaded to the OSF and is now linked in the manuscript, and here:

https://osf.io/r9hm4/?view_only=c768a8f319ed4eb7bf89199a10bbd584

I have a couple of smaller issues that could be addressed or communicated more transparently:

I understand that some of the fieldwork was pre-registered. It appears that Experiments 1 and 3 were pre-registered, but not Experiment 2. Additionally, there were some differences in the pre-registrations. I would appreciate it if these differences could be communicated more transparently. Also, in some instances, analyses deviate from the pre-registrations (e.g., the note on page 10). It could facilitate readers' understanding if the authors could clarify how the results presented in the main body differ from those in the appendix (E1-E2) rather than solely referring to the appendix tables.

RESPONSE:

We have clarified where the analyses deviated from the pre-registration, and also which were the primary analyses (reported in the main text) vs. supplementary analyses (deviations, or exploratory analyses; reported in the appendix).

Experiment 2 was not pre-registered, and this was due to an administrative oversight within the team when preparing the replication study. We now communicate this within the paper.

The paper appears to use a combination of pre-existing stimulus material and material generated or revised for the experiment. In certain instances, it appears that "affectively polarised" words were added to pre-existing material. While I understand that tables in the appendix display all treatment material, I would appreciate additional clarification regarding the fabrication of material for the experiment. Specifically, it would be helpful to see text or a table explaining when and where material was fabricated. Given that norms vary across disciplines, I find it important to note carefully when material was fabricated, the extent to which stimulus material included factually incorrect information, and whether a debriefing took place to make respondents aware of factually incorrect information or manipulations.

RESPONSE:

We have clarified in the method section for each experiment that:

- The headlines (Experiments 1-2) were synthetic, but designed to simulate actual news headlines.
- Stimuli in Experiment 3 were taken from a dataset of real Tweets that included news headlines, with only minimal changes to increase affective salience.

As we had not included synthetic material with factually incorrect information, therefore we did not consider it necessary to include a debrief statement.

The authors could communicate a bit more clearly whether the focus of the paper and the intervention is on reducing engagement with content that is polarising (regardless of its factual accuracy) or specifically on content that is both polarising and factually incorrect.

RESPONSE:

Our focus is indeed on reducing engagement with content that is designed to polarise audiences. Such content can be targeted at individuals who are aligned with either side of an issue, irrespective of the factual accuracy of the content. In our experiments, we did not manipulate content accuracy, only the polarising nature of the stimuli.

Although not the main aim of the paper, we note that polarising tactics are often a characteristic of factually incorrect information (e.g., because it is a desired goal of disinformation) and therefore reducing engagement with affectively polarising content is also likely to reduce engagement with misinformation.

We noted this at the end of the introduction:

“Though much hyper-partisan content contains disinformation, the content we deal with in the current study has no explicit truth value, because factual status is irrelevant to the phenomenon under consideration, which is primarily affective polarisation.”

Finally, I would appreciate it if the authors could demonstrate how they ensure that what they define as "issue polarised" or "affectively polarised" is indeed perceived in that manner by respondents as well.

RESPONSE:

For experiments with synthetic headlines, we tested this with 1,245 participants to check that they were indeed perceived as more non-derogatory (for the issue polarised

headlines) or more derogatory (for the affectively polarised headlines). We report this robustness check of the stimuli in Appendix G.

For experiments with data-driven headlines, they were derived from content where the extent of its affective polarisation had already been determined.

The title of the paper could state that the fieldwork was conducted in the US and the UK.

RESPONSE:

We have amended the title to:

Reducing social media engagement with affectively polarised content via inoculation: Six experiments in the UK and US.

Reviewer #2 (Remarks to the Author):

First of all, thank you very much for being able to review this interesting manuscript. The authors present three research experiments on utilizing inoculation via a technique-based video intervention (experiments 1 and 2) and a text-based intervention (experiment 3). The authors find, in all three samples, that participants indicate less engagement with polarizing content after having been presented with the inoculation intervention vs. a control. However, they also find that the results are not exactly as expected, missing the interaction effects with the specific stimuli, meaning, that the intervention decreased engagement in general, independent of content, for experiments 1 and 2 (UK samples).

RESPONSE:

Thank you for this assessment of our manuscript and the helpful comments and suggestions to improve it. We respond to each of the comments point-by-point below.

Regarding **strengths** of the manuscript, I really appreciated the thorough introduction and theoretical background provided to outline the issue of reducing polarized engagement. This definitely is a very real and relevant problem that researchers should address. Further, the study design was transparent and comprehensively presented (I especially appreciated Figure 1), which allows my in-depth criticism of the methods in the first place, so highly appreciated as well. The main analyses are straight-forward and easy to understand.

RESPONSE:

Thank you for these comments on the strengths of the paper.

Besides the strengths, there are also some caveats, I want to point out. Specifically, I have differentiated them into some **major issues** to comment on and some **minor issues**.

For the major issues:

1. The nature of the outcome measures being indicated intentions of engagement per likelihood rather than actual engagement behavior should be more prominently addressed in the limitations section. In this regard, I very much appreciate the provided evidence on the link between self reported engagement and real engagement. I couldn't find the exact evidence for this in the referenced paper from Guess et al. (2019). Maybe I'm missing something here though? I'd be very glad if the authors could just point me to the respective part in the paper on this. Regarding Mosleh et al. (2020), they indeed find a link between self-report. From my understanding their engagement intention, however, is framed quite differently as a decision to engage or not (in a binary format) rather than asking about sharing or clicking likelihood on a scale, which leaves more room for gut feeling responses (especially suffering from response biases like social desirability or individual reference standards). It might, hence, be meaningful to advance the discussion of this limitation beyond referencing these two papers, clarifying that the measurement procedure in the present studies was different from at least the one used in Mosleh et al. (2020).

RESPONSE:

We have added to the paper 3 additional studies looking at a behavioural measure - text production. In light of these findings (see response to comment below) we now discuss the difference between effects found when measuring intended engagement and other behavioural contexts.

2. The authors did not implement any comprehension checks for the intervention video and no control measures afterwards. I don't want to beat a dead horse in this regard when pointing out the alternative explanation of increased general skepticism in conjunction with more cognitive reflection on the presented headlines as explanation for the results. However, for these results (i.e., overall reduction effect with no further content or stimuli distinctions) in experiments 1 and 2, such an explanation is quite close conceptually and has to be addressed critically. It could be informative to also think about the fact that the interaction with stimuli as expected appears for the text-based intervention (in experiment 3).

RESPONSE:

In light of the reviewer's comments, we introduced a comprehension check in Experiments 4-6, which asked respondents to describe the video they had just

watched.

Both of the above methodological issues might be addressed with another study advanced by two aspects. First, the authors could measure actual behavior. For example, as an outcome measure, participants could be presented with a headline and decide to actually click on it to read more (which would be provided with an additional page providing information connected to the headline). One could even, to include references standard effects (i.e., people being presented with different headlines across a social media session), construct this as a competitive behavior measure, so that two headlines (of different types) are presented simultaneously, and the outcome measure is the ratio of choosing one headline type over the other. Second, in an additional study, the authors could add some comprehension checks for the inoculation intervention to assess if participants have actually understood the central messages from the intervention which would be the prerequisite for the inoculation explanation for the observed effects. Further, adding controls would be important. Depending on what the authors have found for cognitive reflection in experiment 2 (see my 5. major issue commented), they might add this and other controls beyond this to cover alternative explanations of why the implemented intervention video reduced engagement overall.

RESPONSE:

The reviewer raises a good point and these are some interesting suggestions. We conducted three additional experiments with the objective of gathering empirical data showing the effects of the inoculation intervention on a different behavioural measure that goes beyond assessing intentions.

We considered carefully what was the best way to address behaviour. The reviewer presented some very compelling suggestions. However, we also considered Reviewer 1's critique that participants might take the video as instructions for how to engage with simulated social media content presented after the video. As such, we decided to move away from presenting content and used instead a novel measure of self-produced text. By doing so, we obtained a measure of participants' linguistic behaviour (i.e., what they choose to say).

Our additional studies did not yield the expected result: across the three studies, a Bayesian analysis indicated that the intervention had no effect at all on the produced text. However, we believe these provide important findings on the limitations of inoculation: while it was effective based on test items that are commonly used, the generalisability of the effect to other contexts such as written text production remains uncertain.

3. In experiment 1, the authors report a third engagement measure of emoji reactions, which they then left out for experiment 2. No results are reported for it (at least, I

cannot find them), focusing on the combined score of sharing and clicking likelihood in experiment 1. For transparency, it would be important to also report this measure's results and discuss it like the other two.

RESPONSE:

We report the results of the emoji reactions for Experiment 1 in the appendix (F). This was pre-registered for Experiment 1 as an exploratory analysis so is not part of the primary focus for the paper. To make Appendix F clearer, we have noted that the "reactions" reported here are the emoji reactions.

4. From experiment 1 to experiment 2, the authors switch for their central outcome measures from a 4-point Likert scale to a 10-point Likert scale which is a substantial jump. It would be meaningful to provide an explanation for why this was done (I'm assuming to increase sensitivity of the likelihood measure? But then why the jump from 4 to 10 exactly?).

RESPONSE:

The reviewer is correct that sensitivity of the measure was the reason for the change. We have explained this decision in the manuscript.

The four-point Likert scale employed in Experiment 1 was substituted for a 10 point linear scale based on work that suggests that using 10-point scales increases the variation in responding over 5 and 7-point scales (Dawes, 2008 - doi: <https://doi.org/10.1177/147078530805000106>).

5. In experiment 2, the authors add a cognitive reflection measure. However, I don't find the results on cognitive reflection, how it predicts engagement or interacts with the inoculation intervention. If this cognitive reflection predicts overall engagement and is even increased by the inoculation intervention, that would corroborate the alternative explanation of general cognitive reflection or skepticism to reduce the overall engagement. It would be important to provide these results on cognitive reflection for experiment 2.

RESPONSE:

We found no effect of cognitive reflection as a covariate. We have noted this now in the results and discussion section for experiment 2.

6. The authors find the effect pattern as predicted for experiment 3 in contrast to experiments 1 and 2. As far as I understand, the central explanation provided is that the samples are different, namely UK vs. US participants. My best speculation of the reason of the shifting effect pattern is the type of intervention employed. Namely for experiment 3 the authors switched from a video-based to a text-based intervention.

This should be prominently outlined in the interpretation of the results and further discussed in the limitations and future research section. It might point to a crucial realization of differences between video-based and text-based approaches of inoculation techniques that moves beyond the explanation via countries.

RESPONSE:

We conducted three additional experiments using the video intervention in the US, on the same topic as Experiment 3. As this produced a different effect, it seems likely that the differences across the experiments are more to do with topic, response measures, and sample rather than intervention type.

7. For the interpretation of the results, the authors note that (p. 15): “The UK sample is especially strong, being composed only of those who report having previously engaged in sharing political content on social media. This makes the efficacy of our intervention all the more compelling [...]” However, isn’t it then especially critical that the effect pattern for experiment 1 is in some points not conclusive and this inconclusiveness is replicated in a second independent experiment 2; where both experiments are based on UK samples?

RESPONSE:

We have rewritten this section in light of the additional experiments we conducted.

Minor issues/questions:

1. I don't fully understand why experiment 2, introduced as the replication of experiment 1, is the only study not preregistered. If not all studies are preregistered, readers might expect this to be the other way around. The authors might just add a clarification (even if a practical one) of why this is the case (i.e., study 2 not being preregistered).

RESPONSE:

Experiment 2 was not pre-registered due to an administrative oversight within the team when preparing the replication study. We now communicate this within the paper.

2. A probably more subjective note is that, from time to time, I was a bit confused by what was found in the Appendix. I understand and appreciate the attempt to be as brief as possible in the main parts. For the results specifically, however, I'd argue that it is well within the range of the main text to present results for each experiment that can cover one to two pages. For example, I don't think that the clicking and sharing likelihood has to be combined for brevity in the experiments 1 and 2. Rather the authors can just present the results on both (and in experiment 1 also the emoji reaction) in the main text and link their tables in the main text as well.

RESPONSE:

To adhere to the journal's word limit for articles (approximately 5000 words), we had to make some reporting decisions to enable conciseness in presentation. We have made these decisions clearer for what is presented in which section (main results or appendix) clearer. In brief, primary pre-registered analyses are reported in the main text, while secondary pre-registered analyses (e.g., mediation) and exploratory analyses (e.g., emoji reactions) are reported in Appendix E & F.

In summary, even if this doesn't directly come across due to the set of issues noted, I really like the manuscript (I just find this such critical research that I want to do my best to support this as much as possible) and I think publication in Communications Psychology would be suitable. I would also not argue that an additional study with the recommended advancements is necessary per se, I leave this up to the editor. I would, however, argue that major revisions are necessary regarding explanations of methodological decisions (especially switches between experiments like from video- to text-based inoculation) and presenting the results of specific main outcomes (i.e., emoji reaction) and control variables that are important for the interpretation of an inoculation effect (e.g., cognitive reflection). Additionally, I think advancing the limitations section might be super interesting for any followup research. I understand the present manuscript as an opportunity to provide an informative basis (with very high-quality samples) for many research yet to come, which includes insights on what works and what doesn't work regarding inoculation.

Thank you very much, once again, for being able to review this manuscript, I'm looking very much forward to a revised version!

RESPONSE:

Thank you for your helpful suggestions.

Reviewer #3 (Remarks to the Author):

1) Lines 57-58, you write that: "Creators of such content have varied objectives, but often seek to influence and reduce trust in democratic processes by increasing group polarisation". I understand that this might be true in some cases, but it's also possible (and I would argue more commonly) that the dissemination of misinformation is a by-product of the financial incentives of sharing sensationalist or partisan news.

RESPONSE:

This is a good point. We have amended this sentence to:

"Creators of such content have varied objectives, including monetisation of the sharing of sensationalist or partisan news, but often there is an incentive to influence and reduce trust in democratic processes by increasing group polarisation"

2) Line 66: capitalize Twitter and Facebook.

RESPONSE:

We have fixed this.

3) I really like how you summarized the debate on nudges on the paragraphs between the lines 87-108.

RESPONSE:

Thank you!

4) Lines 141-144 “Osmundsen et al. (2021) focused on all types of political news sharing in their analysis, finding that out-group derogating content in news headlines did not just predict sharing of fake news by out-group derogating content, but also the sharing of news from any outlet.” As someone who was not familiar with this work from Osmundsen, I did not understand the key takeaway based on the sentence. Did out-group derogating content predict sharing news from any outlet? How does a type of content predict sharing of all types of content? Some clarification might help here.

RESPONSE:

We have clarified this sentence:

Osmundsen et al. (2021) focused on all types of political news sharing in their analysis, finding that out-group derogating content in news headlines predicted sharing of *any* news content, not just fake news.

5) Line 397: You mention YouGov exclusion criteria for speeders. Please specify what criterion they use (e.g., how many SDs above/below mean?)

RESPONSE:

We have amended to clarify the exclusion criterion in the text:

Quality of responses was ensured by YouGov, which automatically excluded speeders (a completion time two or more standard deviations below the mean) and suspected bots (using geo-tagging and straight-line checks).

6) Throughout the paper, you only have one example for the headlines for “remainers” and “leavers” I’d like to see examples of the derogatory vs non-derogatory headlines as

well. That would be suitable for the methods section, as you're describing the stimuli, or for a figure/table.

RESPONSE:

We refer to one example within the main text, but the exact text of all stimuli headlines is presented in the Appendix G.

7) As I started reading the paper, I kept asking myself about how much the intention of sharing translates into real world sharing. I see that you addressed this question by the end of the discussion (lines 354-357). This information should be earlier in the paper, perhaps by the end of introduction, when you're outlining the studies. I would also consider appropriate to report the magnitude of the correlation (e.g., "previous experiments employing similar designs found that self-reported engagement with stimuli was strongly associated with real world engagement on social media, with correlations ranging from x to y").

RESPONSE: We have added to our introduction of the experiments with self-reported engagement:

Previous experiments have found self-reported engagement (e.g., intention to share) correlated with actual sharing behaviour ($r = .44$).

8) The way the stimuli are categorized is confusing to me. It seems like different sections of the paper use different labels. You mention Derogatory VS non-derogatory, issue polarized VS affectively polarized, scapegoating VS ad-hominem VS emotional language headlines. In your results section, it's sometimes hard to figure out what analysis you're referring to exactly. Part of the confusion might be because some of these terms are used interchangeably, but I couldn't tell with certainty if that was the case. For instance, study 1 reports a) reduced overall sharing, b) participants engaged more with non-derogatory content, and c) the interaction between intervention type and headline type (issue polarized VS affectively polarized) was not significant. In study 2, you say that you replicated the results of study 1. However, the first result you report is that "The inoculation significantly reduced engagement with affectively polarised political content". What finding from study 1 is this replicating? a, b, or c? It seems like you're referring to result (c), but it didn't make sense because study 1 did not find that it was significant.

RESPONSE:

We have clarified the wording in these sections as follows:

- (1) We now use a single, consistent terminology for the manipulation (affectively polarised)
- (2) We explain that the manipulation of affective polarisation in the headlines was implemented using three different polarisation techniques.
- (3) There was an error in the first report from study 2, which was the main effect in the ANOVA. The first result is that inoculation reduced overall engagement with both types of polarised content.

Sometimes you refer to your conditions as derogatory VS non-derogatory and sometimes as affectively polarized VS issue polarized. If they are used interchangeably, please choose only one, as two names for the same categorization make readers confused. I think derogatory and non-derogatory are more intuitive terms.

RESPONSE:

We have checked to make sure our terminology is consistent throughout.

9) In lines 189 – 193, you describe the strategies for inoculation as top-down/bottom-up. These strategy names could more intuitively be called rationality-based and emotion-based, or something similar. Top-down and bottom-up might make sense for the readers from the brain sciences, but the audience for this paper seems to be broader, so I'd avoid that.

RESPONSE:

We have clarified the wording of this paragraph, as here we refer to the approach of delivering the inoculation as being theoretically driven vs. data driven (rather than brain processes). It now reads:

“By doing so, we pursued a data-driven approach to inoculation in place of the theory-driven approach taken in Experiment 1. A large body of research supports the efficacy of theory-driven inoculation techniques against rhetorical devices of argumentation (e.g., incoherence, ad-hominem, scapegoating, etc.). Here we propose a ‘data-driven’ technique: inoculating against the use of linguistic features that have been shown to reflect affective polarization”

10) Throughout your discussion session, you mention a few times that your intervention was successful to decrease the amount of affectively polarized content. However, that is unequivocally true only for experiment 3. In studies 1 and 2, the intervention decreased all types of sharing, which is not the same as saying that your intervention worked as hypothesized. Please explore in your discussion what could be some of the negative consequences of such intervention decreasing overall sharing behavior. What could happen if people decreased sharing of non-polarized stimuli? Is that something desirable?

RESPONSE:

We have rewritten the discussion as we conducted and now report three additional experiments that offer more empirical data on the limitations of the intervention.

11) Lines 354-357: “On the question of ecological validity, previous experiments employing similar designs found that self-reported engagement with stimuli was strongly correlated with real world engagement on social media (Guess et al., 2019; Mosleh, Pennycook, & Rand, 2020).” Throughout the paper, the question about ecological validity was popping in my mind, if it was stated earlier in the paper, I might have dismissed some skepticism earlier. This information should be earlier in the paper, perhaps by the end of introduction or beginning of results, when you’re outlining the study. I would also consider appropriate to report the magnitude of the correlation (e.g., “previous experiments employing similar designs found that self-reported engagement with stimuli was strongly associated with real world engagement on social media, with correlations ranging from x to y”).

RESPONSE:

We have addressed this in our response to comment 7.

12) Minor note: First sentence of abstract feels a little out of place because the focus is on the agent “malevolent actors” rather than the action “disruption of democracies”. This is a matter of style, but I would rewrite it to avoid too much attention to the malevolent actors. (On a first read it sounds a little sensationalist).

RESPONSE:

We have amended this to:

“The generation and distribution of hyper-partisan content on social media has gained millions of exposure across platforms, often allowing malevolent actors to influence and disrupt democracies.”

Reviewer #2 (Remarks to the Author):

I thank the reviewers for addressing all of my comments. Given these clarifications and content-additions and the immense empirical additions in the revised manuscript with clearer results that provide a more nuanced understanding of the limits of the inoculation effect (i.e., on actual behavior), I suggest to accept the manuscript.

Thank you for your valuable feedback!

Reviewer #3 (Remarks to the Author):

I appreciate that the authors have addressed all my previous comments in the revised manuscript.

The addition of three new studies to respond to other reviewers' feedback is noted, and I commend the authors for employing the DDR approach to analyze the new data.

However, I have concerns regarding the implementation of the writing task in Studies 4 and 6, especially in relation to addressing the feedback about measuring real engagement. The writing task, while informative, does not seem to directly tackle the issue of genuine engagement with social media interactions. This task does not mimic the interactive nature of social media environments where participants engage with others' content. Additionally, the behavior assessed in the writing task appears tangential to the main independent variable and the broader discussion within the manuscript. If the intervention focuses on how individuals react to online stimuli, it is unclear why producing a written text on a political topic, such as personal opinions on abortion, would be influenced by the experimental manipulation.

Study 5 presents a task more aligned with the dependent variable of interest—participants' reactions to content they might encounter online, an online post. However, it remains unclear how this task better captures real engagement compared to the measures used in Studies 1-3, as it is not capturing the dynamics of online interaction. Furthermore, several potential confounding factors could have influenced participants' responses. For instance, participants might temper their emotional reactions knowing that: (a) their responses are being reviewed by researchers, and (b) their written input will not be publicly posted or read by the original comment's author or an online audience.

Finally, I believe there are fundamental differences between the cognitive demands of writing a response and the simpler actions of liking or sharing content online. The last three studies appear to shift between the concepts of content consumption and content creation. The effort required to write a detailed opinion is significantly greater than the cognitive load associated with engaging through likes or shares. Even if the inoculation intervention proves effective for these more passive forms of engagement, I do not think it is fair to assume that it would generalize to content creation. While these behaviors may be interconnected, the rationale for

using content creation as a dependent variable is unclear, given the paper's primary focus on content consumption and engagement.

In conclusion, while I appreciate the authors' thorough work, I find the manuscript to have been more cohesive with the initial three studies only. I leave the decision to your discretion but encourage careful consideration of the alignment between the research questions and the dependent variables assessed in the later studies.

We thank the reviewer for their thoughtful feedback and agree with their comments. We have now added the following to the limitations section:

“A fundamental limitation concerns the distinct types of engagement across our studies. While Experiments 1-3 examined low-effort engagement (clicking, sharing), Experiments 4-6 investigated content creation through writing tasks. These represent fundamentally different cognitive processes, which may explain why our intervention was effective only for simpler forms of engagement. Additionally, our writing tasks did not fully capture the interactive nature of social media environments, with participants potentially moderating their language due to researcher oversight and the absence of a real online audience.”

And “Future research should focus on developing and testing interventions that can address both content consumption and creation in more naturalistic social media settings. Such work will be crucial for understanding how inoculation effects might translate to real-world behavior across different forms of social media engagement.”